# Rapid Truck Loading for Efficient Feedstock Logistics

Robert "Bobby" Grisso [1,*] , John Cundiff [1] and Subhash C. Sarin [2]

1 Biological Systems Engineering, Virginia Tech, Blacksburg, VA 24061, USA; jcundiff@vt.edu
2 Industrial and Systems Engineering, Virginia Tech, Blacksburg, VA 24061, USA; sarins@vt.edu
* Correspondence: rgrisso@vt.edu; Tel.: +1-540-231-1980

**Abstract:** A multi-bale handling unit offers an advantage for the efficient hauling of round bales. Two empty racks on trailers are left at a satellite storage location for loading while a truck tractor delivers two loaded racks to the biorefinery, thus uncoupling the loading and hauling operations and increasing the efficiency of both. The projected 10 min trailer exchange time equals the projected 10 min unload time at the biorefinery achieved by lifting off the two full racks and replacing them with two empties, a technology adapted from the container shipping industry. A concept is presented for a bale loader that latches onto the rack/trailer and loads bales into the bottom tier chambers. This machine will load 10 bales into the rack on the front trailer by attaching on to the front of the trailer and 10 bales into the rear trailer by attaching onto the rear. A telehandler removes bales from single-layer storage and places them in the bale loader to load the bottom tier compartments. The top tier compartments are loaded directly from the top. Expectations are that an experienced operator can average 9 loads in a 10 h workday, and load-out cost is estimated as 3.61 USD/Mg, assuming the average achieved load-out productivity over annual operation is 60% of optimum productivity (24 Mg/h) equal to 14.4 Mg/h. Cost increases to 4.81 USD/Mg when the productivity factor drops to 45%, and cost is 3.09 USD/Mg for a factor of 70%.

**Keywords:** biomass; biomass logistics; hauling costs; in-field hauling; location allocation; management systems; satellite storage locations; transportation





## 1. Introduction

This manuscript advocates a concept for rapid loading of round bales into a multi-bale handling unit at satellite storage locations (SSLs). The analysis is specific to a biorefinery located in the Piedmont. The Piedmont, a physiographic region covering a significant part of five Southeastern States (VA, NC, SC, GA, and AL), has abundant rainfall, long growing seasons, and adapted plant species can support high levels of feedstock production. Of equal importance, the region has significant available land that can be converted to feedstock production with negligible impact on food production, or concern over indirect land use change.

There are three premises for the concept presented here.

1.   The feedstock is a perennial grass, specifically switchgrass (*Panicum virgatum L.*). This biomass will be harvested in 5x4 round bales (5 ft dia. X 4 ft wide) and stored in single layer ambient storage at SSLs. Hereafter the term "bale" refers to these round bales.
2.   Feedstock contractors will grow, harvest, and place bales in SSLs. The biorefinery takes ownership when the biomass is removed from this storage.
3.   Companies with delivery contracts will load-out biomass from an SSL, and companies having year-round contracts with the biorefinery will do the subsequent highway hauling to a biorefinery to meet a weekly demand for annual operation. Hauling of bales with a multi-bale handling unit was discussed by Grisso et al. [1]. The analysis presented here is a continuation of this previous work and focuses on the equipment to load the multi-bale handling unit, a 20-bale rack. The key issue with any short-haul contract is the load and unload times. These times must be minimized to maximize

truck productivity (Mg/d). Cost to operate tractor-trailer trucks (USD/d) is well established by the 200,000+ trucking companies in the USA. Trucking cost (USD/Mg) is minimized when productivity (Mg/d) is maximized.

## 2. Justification

The total time during a day when the truck is waiting to be loaded, and the total time the SSL load-out crew is waiting for an empty truck, is reduced when any available truck can be dispatched to any SSL where a load is available. (This management plan is defined as central control of feedstock hauling, meaning that a "Feedstock Manager" at the biorefinery is in communication will all trucks and SSL load-out operations, and directs these operations from a central location.) Aguayo et al. [2] reported an analysis of corn stover delivery from real-life "roadside" storage locations in the Midwest (USA). They used a fixed load and unload time for each load (no delays) and showed that average delivered cost across the entire harvest could be reduced by 20% using central control of hauling.

Load-out is defined as all operations at an SSL required to load biomass for highway hauling. Mobile equipment is defined as load-out equipment that finishes load-out at one SSL and then moves to the next. Mobile equipment for load-out of biomass at SSLs has been investigated by Judd et al. [3]. They have shown that the utilization of mobile equipment results in average savings of 14.8% over equipment placed permanently at SSLs. A judicious selection of SSLs relative to production fields is also important to achieve minimal logistics cost. For instance, they showed a reduction of 39.3% and 47.7% in transportation distance from the production fields to the SSLs, in comparison to heuristic (quick, common sense) solutions provided by Resop et al. [4] for the datasets pertaining to production fields within a 32- and 48-km radius around the biorefinery, respectively. Judd et al. [3] also considered three different types of SSL load-out equipment (side-loading rack, rear-loading rack, and a densification system), and reported that the use of a side-loading rack results in average savings of 21.3% over the densification system. In another study, Aguayo et al. [5] have shown that a rapid system to unload racks at the bio-energy plant can achieve significant savings.

A variation in the amount of biomass harvested and delivered to SSLs each month (harvest schedules) can contribute a significant increment to total cost. Therefore, it is prudent to determine the most likely harvest schedule for a cost-effective supply chain design. To ensure a continuous operation of the biorefinery in the face of uncertainty, it is essential to hold a suitable amount of inventory at each SSL and/or at the biorefinery. The amount of biomass placed at these locations depends on the level of uncertainty encountered. Another strategy is to control the harvest schedule by giving economic incentives to farmers so that they will harvest and place in SSLs the amount of biomass contracted for each month. This will help in reducing the impact of harvest schedules on supply chain decisions. The inclusion of extra SSLs in the supply chain design can also serve the role of a reserve stock to deal with shortages, biomass loss resulting from degradation, unexpected yield loss, or changes in harvest schedules.

### 2.1. Business Plan for Biorefinery—Central Control of Feedstock Delivery

Grisso et al. [1] described a rack holding 20 bales, and the concept called for this rack to be hauled on a trailer with the attachment points and handling features used by the container shipping industry [6]. Units with two trailers hooked in tandem are legal in all 50 states, and this is defined as a truckload (40 bales) for this analysis. Simply stated, the concept applies mature technology developed for the container shipping industry to the feedstock logistics problem.

Using an example to orient the reader, calculations are done for a biorefinery averaging a bale-a-minute for 24/7 operation. This biorefinery will consume 60(24)(7) = 10,080 bales/wk, and at 40 bales per load, the demand is 252 loads/wk. If loads are received 6 d/wk, the average delivery is 42 loads/d. If the haul day is 12 h, the receiving operation at the biorefinery must average 42 loads in 12 h or 3.5 loads/h. The feedstock must flow into,

and out of, biorefinery storage to maintain continuous operation. The biorefinery structure required to receive truckloads of feedstock, place the racks in short-term storage, and retrieve the racks for continuous operations is defined as the "Receiving Facility."

The receiving facility operation is a critical component of the logistics chain needed to deliver feedstock for continuous operation, because it defines the truck unload time. Single-bale unloading and placement into biorefinery storage is not a practical option. The labor cost (USD/Mg) is too high; also, truck cost is increased because the unload time is increased, and subsequent truck productivity (Mg/d) is reduced. The goal set for the concept presented here is a 10-min unload time. The unload time is defined as the time allotted to weigh in a truck, sample for quality, unload, and weigh out the truck. To achieve this goal, the concept presented here uses the multi-bale handling unit described by Grisso et al. [1].

In the Southeast, cotton growers' contract with a gin to receive and gin their cotton. The same is true with the sugar cane industry in Louisiana and Texas. Today, tobacco is grown under contract. No one in these industries delivers their crop to a processor to see what spot price is being paid on a given day. The authors state directly that we do not believe that the uncertainty of a spot market will ever be a viable option for a biorefinery. Random deliveries could result in 10 loads one day and 50 loads the next. If the Receiving Facility is designed for 42 loads/d, the feedstock logistics chain will operate cost effectively when 42 loads, plus or minus some contingency, are received each day. Truck cost, and receiving facility cost, are both optimized with central control of highway hauling. The need for central control of the logistics is well defined in many different industries. Examples are municipal solid waste pickup in towns and cities, deliveries from a central warehouse to a chain of stores, and many manufacturing plants that need materials delivered on a given schedule to keep their assembly lines moving.

### 2.1.1. Rack Design

The current rack and trailer prototypes were built by Sea Box [6] (Figure 1). The rack has four chambers, two on the bottom tier and two on the top tier. Each chamber holds 5 bales for a total of 20 per rack.

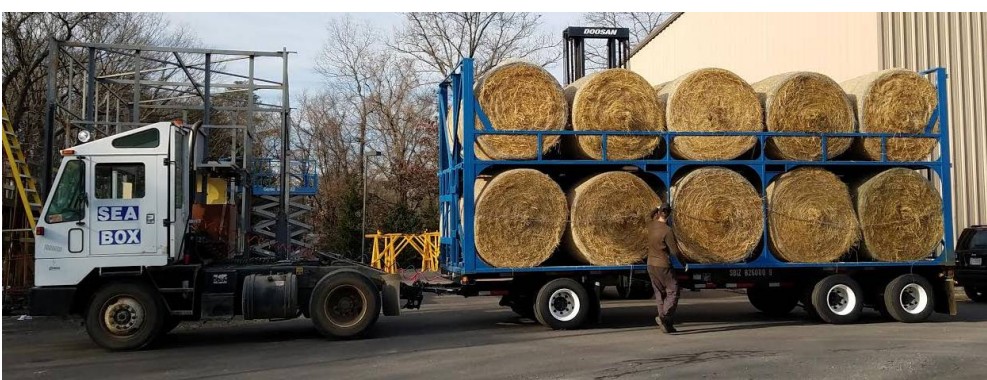

**Figure 1.** Prototype rack and trailer (Sea Box [6]).

### 2.1.2. Receiving Facility Operations

To unload a truck, a forklift (16 Mg capacity) removes the loaded racks and replaces them with empty racks. The operation of this forklift, and the efficiency of this operation, is well known by those who manage the unloading and handling of shipping containers at a depot. Container shipping is a mature (dating back to the mid-1950s) and well-established industry. The handling of bales in a 20-bale unit, and the resulting improvement in the productivity of receiving facility operations, is the major advantage of the rack concept.

The concept envisions that bales will stay in the rack until processed. For this analysis, all short-term storage in the receiving facility will be bales in racks. When a rack is removed from a truck entering the receiving facility, it can be processed immediately, or placed in

short-term storage for nighttime, or weekend, operation. Figure 2 shows filled racks in storage (right-side) and empty racks (left-side) ready for transport back to an SSL, thus it is a "snap shot" of a condition midway through nighttime, or weekend operation. Not shown is the 16-Mg forklift used to place full racks in the rack unloader and empty racks in storage.

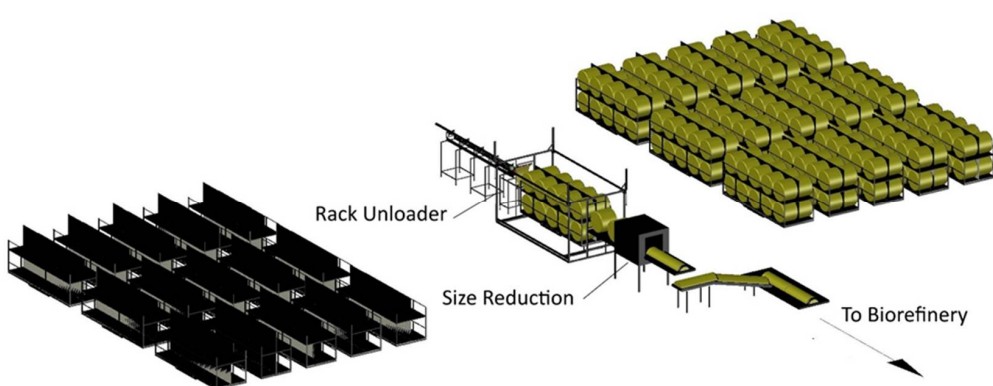

**Figure 2.** Concept for unloading and storing racks at biorefinery.

A prototype rack unloader (Figure 3) has been built [7]. The 16-Mg forklift places a rack in the rack unloader, where it is unloaded to establish a single line of 20 bales on the feed conveyor delivering biomass into the biorefinery for 24/7 operation.

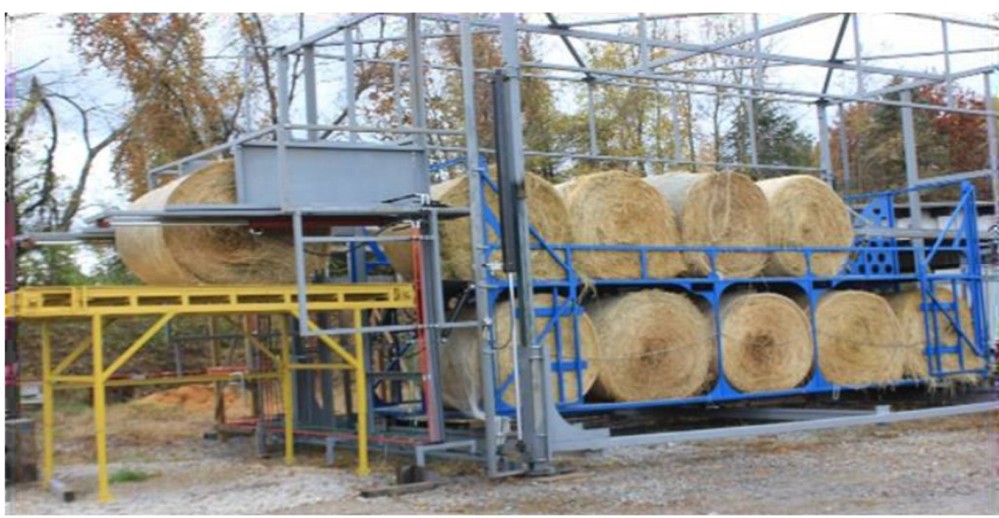

**Figure 3.** Rack unloader shown unloading bales from top tier chambers.

The prototype rack unloader works as follows. The rack is shifted such that one of the two top tier chambers aligns with the conveyor. A push-plate extends and pushes these 5 bales onto the conveyor (Figure 3, near top tier chamber). Then, the rack is shifted to push bales from the other top tier chamber. Next, the rack is lifted and one of the bottom tier chambers is aligned and the bales pushed out. In like manner, the second bottom tier chamber is emptied. This procedure produces a line of 20 bales on the conveyor for delivery to the size reduction equipment.

Initial testing showed that the unloading of the top tier chambers was routine, and bales were extracted without issue. The bales in the bottom tier chambers tended to shift during transit and one, or more, bales would hang on one of the vertical side-supports. It was then impossible to push out the line of bales.

A redesign of the rack unloader to provide for correction of multiple bale misalignments during unloading was considered and rejected. The decision was made, and pre-

sented here, to change the rack design, install a side rail for the bottom tier, like the top tier, and load the bottom tier compartments from the rack end. This design will correct any bale misalignments at the SSL, maintain this alignment during transport, and a "trouble free" loaded rack is delivered to the biorefinery.

Every bale in the handling unit (rack) is handled once. It is in a defined position from the time it leaves the SSL to the time it enters the biorefinery. Minimizing bale handling is a very important aspect of the rack concept. If bales are stored in single-layer ambient storage in SSLs for 6 months, or more, then minimum handling in the logistics chain is a requirement, not just a cost-reduction option.

## 3. Design Considerations

### 3.1. Load-Out and Hauling Operations

A 10-min rack load goal was assumed for this study. This goal cannot be achieved if the truck is parked for loading. Uncoupling the rack loading and rack hauling operations is essential.

The concept envisions that tandem trailers with empty racks will be dropped at the SSL and the truck driver will hook to tandem loaded trailers for the return trip to the biorefinery (Figure 4). This procedure can be accomplished in 10 min, thus it meets the 10-min load time criteria. Note, the prototype trailer shown in Figure 2 is towed with a pintle hook hitch. Hitching is quickly done using a back-up camera in the truck cab.

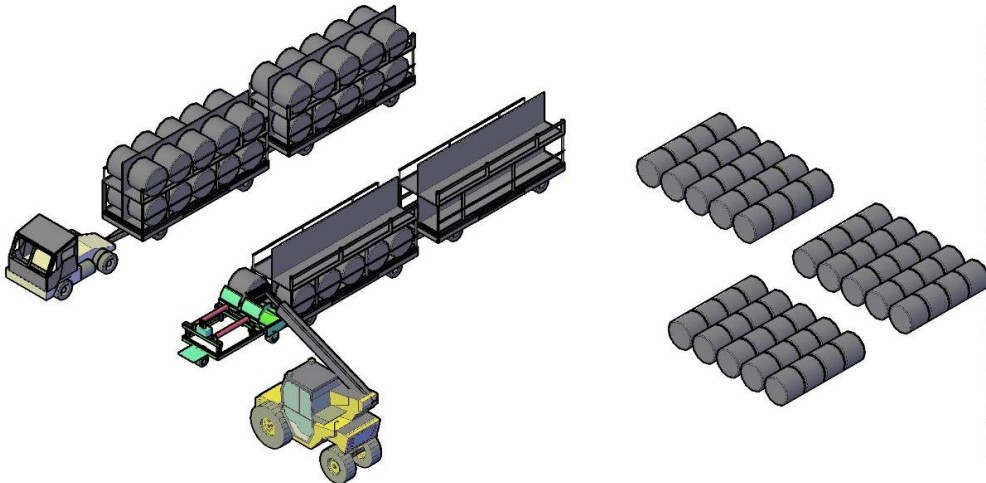

**Figure 4.** Concept for bale loading at SSL showing an empty rack trailer being filled and a loaded trailer leaving for transport to biorefinery.

Racks remain on the trailer for bale loading. The concept requires that each truck tractor will have two, or more, tandem trailer units assigned.

There are at least two ways to envision the hauling contracts.

1. Option 1 envisions a single contract where the contractor owns the required load-out equipment and owns the number of trucks to deliver the biomass they load. A single delivery fee (USD/Mg) is paid for feedstock weighed in at the biorefinery. The contractor schedules both the loading and hauling, thus there is no provision for hour-by-hour (central) control by the biorefinery.
2. Option 2 envisions a contract that pays a fee (USD/Mg) for loading at the SSL, and a separate fee to a company operating truck tractors to pull the trailers. For this option, the control of individual trucks (any truck can be sent to any SSL where a load is waiting) is arranged by the "Feedstock Manager" at the biorefinery. The option opens up an opportunity for a small business (an individual with only one truck tractor) to negotiate a contract to pull trailers.

Both options require the biorefinery to own and maintain the racks and trailers. With central control, the total wait time for a truck at a load-out operation can be minimized. In like manner, the total time an individual truck waits for a load can be minimized. We selected Option 2, as this option provides for continuous control by the biorefinery, thus it gives the potential for the lowest average delivered cost for feedstock. Productivity of both operations is increased.

### 3.1.1. Current Load-Out Operations at SSL

The current rack design uses a telehandler to load bales into the bottom tier from the side, 5 bales from the right side and 5 bales from the left side. The top of the rack is open so the top tier bales are loaded by lifting them and lowering them down into position.

### 3.1.2. New Design for Bale Loader

The end-loading rack design has a side rail for the bottom tier chambers like the side rail for the top tier chambers. These rails ensure the bales cannot become misaligned during transport. End loading requires a machine, hereafter referred to as a "Bale Loader", that emulates the forklift that is carried on the back of a delivery truck, typically a truck delivering building materials. This machine, along with the telehandler, is stationed at the SSL. When a truck arrives and drops a trailer set, the bale loader is driven into position, locked onto the rack trailer, and thereafter operated remotely by the telehandler operator. (A single operator will perform all operations at the SSL.) The bale loader will have two push plates, one for each of the lower tier chambers (Figure 5a). The telehandler operator will place a bale on one side and activate the push-plate cylinder. This cylinder will cycle, extend and retract, to push the bale into the chamber (Figure 5b). When the next bale is inserted, it moves the previous bale forward. This operation will be repeated until all 5 bales are loaded and the chamber filled. (The maximum force requirement occurs when the fifth bale is inserted because the previous four bales are moved forward.) Then, the telehandler operator will load bales to fill the second lower tier chamber (Figure 5c).

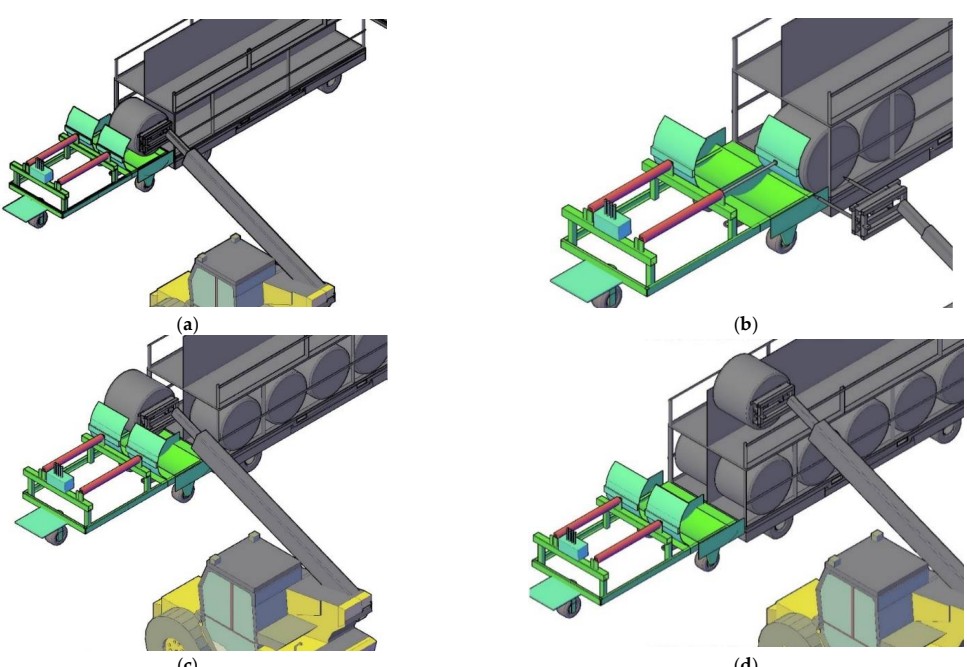

**Figure 5.** (**a**) Telehandler placing bale into bale loader used to load bales into bottom tier chamber of rack, (**b**) Bale loader shown inserting a second bale into the bottom tier chamber of rack. (**c**) Filling back lower-tier chamber. (**d**) Concept for telehandler loading bales in top tier chambers.

The floors of the chambers of the current prototype rack are sheet metal with ridges about 25 cm apart. Bales readily slide on this surface. (The push plates are designed to prevent rolling.) The amount of force required to push a line of 5 bales from one of the rack chambers was measured during the tests of the prototype rack unloader (Figure 3). Based on limited tests, the maximum force to push out the top tier bales was 10.2 kN. The force requirement to push the last bale into the lower tier chamber can be higher than 10.2 kN because there is potential for a bale, or bales, to ride up and contact the roof of the chamber and thus increase the slide resistance. As an example of a possible design, suppose a 125 mm bore cylinder is selected and the estimated maximum force is 30 kN. The pressure required is 25 bar. A pressure relief valve setting of 200 bar is routine for a circuit like this, and at this pressure the maximum force is 245 kN, over 8 times the estimated maximum required. Generating the required force to fill the bottom tier compartment is a routine hydraulic design problem.

The bale loader has sufficient clearance to drive over the top of the tongue without interference and latch in place on the lead trailer. This means the lead trailer will be loaded from the front-end while hitched in tandem (Figure 4). The rack loader is then repositioned to load the rear trailer from the rear-end. The new end-load rack design is symmetric with the same bale loader attachment points at each end.

Bales are loaded into the top tier compartments as shown in Figure 5d. The telehandler places each of these bales into position along the length of the rack. The open top rack design allows for this loading. The current rack design (Figure 3) does have reinforced corner posts so that the racks can be stored two high, if desired, to minimize the central storage area required at the biorefinery.

The functionality of the bale loader is straight forward. Wireless control of the maneuvering functions to latch it onto the trailer is not needed. The telehandler operator will mount the bale-loader and use manual controls on the bale loader to maneuver it into position and lock it to the trailer. A solenoid-actuated directional control valve, operated from the seat of the telehandler, is used to cycle the hydraulic cylinder. A wireless electronic control box (widely available and used in many industrial applications) is mounted on the directional control valve, and a signal from this wireless control in the cab of the telehandler cycles the push cylinder. It is emphasized, one operator does the entire bale-loading operation.

## 4. Discussion

### 4.1. Operation Plan for SSL Load-Out

For this analysis, the goal for the rack-loading operation at the SSL, not yet verified with experimental data, is to load 20 bales into a rack in 20 min. This is optimistic, but believed to be doable with an experienced operator, and an SSL layout which provides for easy access to both sides of the rack trailers. In a workday with 10 productive hours (600 min), and no delays, the operation can theoretically load 600/20 = 30 racks, or 15 truckloads. It is not suggested that this can be achieved in a production setting.

Assuming a bale weight of 400 kg at 15% moisture content, the mass of a 40-bale truckload is:

$$\frac{40 \text{ bales} \left(400 \frac{\text{kg}}{\text{bale}}\right)}{1000 \frac{\text{kg}}{\text{Mg}}} = 16 \text{ Mg/truckload} \tag{1}$$

We assume that a commercial load-out operation will average 60% of theoretical capacity, thus, the per-day loading will be 0.60 (15 loads/d), or 9 loads/d. In a 10-h workday averaging 9 loads/d, the achieved SSL load-out capacity is 14.4 Mg/h.

The achieved average productivity across a 48-week season could be as low as 0.45, if wait times are greater for a set of empty racks to be delivered to the SSL. With this productivity, the average loads/d is 6.75 and the achieved capacity is 10.8 Mg/h. In like manner, if the achieved productivity averages 0.7 (optimistic for a 48-week season), the loads/d is 10.5, and the achieved capacity is 16.8 Mg/h.

### 4.2. Cost for Rack Loading

An estimated cost for operation of the bale loader, assuming equivalence with a commercial piggy-back forklift, is given in Appendix A. Load-out and hauling will be done year-round, thus the cost analysis is based on 48 wk/y operation. No labor cost is included because only one operator is used, and this labor cost is included in the telehandler cost. Using the USD/h cost from Appendix A, an average bale loader cost for the 0.6 average productivity is estimated to be:

$$\frac{10.32 \text{ USD/h}}{14.4 \text{ Mg/h}} = 0.72 \frac{\text{USD}}{\text{Mg}} \tag{2}$$

The same costing procedure as shown in Appendix A is calculated for a telehandler. Using an estimated purchase price (84,055 USD), design life (10,000 h), and repair and maintenance factor (2.67 USD/h), the operation cost is 41.55 USD/h for the telehandler, which includes a labor cost (25 USD/h) for the operator. The cost of the telehandler, 0.6 average productivity, is:

$$\frac{41.55 \text{ USD/h}}{14.4 \text{ Mg/h}} = 2.89 \frac{\text{USD}}{\text{Mg}} \tag{3}$$

Total cost for load-out operations is then 0.72 (bale loader) plus 2.89 (telehandler) combined for 3.61 USD/Mg. This cost does not include the cost of a service truck needed to support operation of multiple SSL load-out operations, or a mobilization cost to move load-out equipment (telehandler and bale loader) from one SSL to the next. A contractor with multiple SSL load-out operations will have an equipment hauler on standby to move equipment as needed.

Using the same procedure for the 0.45 average achieved productivity, the cost for load-out operations is 4.81 USD/Mg. The cost is 3.09 USD/Mg for the 0.70 average achieved productivity factor.

### 4.3. Biorefinery Operational Plan

A key question is, will a biorefinery consider a business plan where all at-plant inventory is storing bales-in-racks? Other herbaceous biomass industries, specifically the cotton (2.2-m diameter round bales of seed cotton) and sugarcane (bins similar size as 20-bale rack) industries, maintain sufficient inventory in their large handling units to support nighttime and weekend operation, so commercial models are available.

Specific questions are:

1. Does the biorefinery desire to accumulate sufficient inventory in loaded racks at the end of the 6-day haul week to supply operations over the weekend?
2. Should empty rack trailer sets be pre-positioned over the weekend to the SSLs assigned for load-out Monday morning? This requires extra operating time by selected truck tractors, however, it insures load-out can begin without delay.
3. Should a multi-day biorefinery inventory be maintained in racks for operation when hauling is delayed due to inclement weather? There will be winter days in the Piedmont, when roads are impassable due to ice and snow. Heavy rain will also delay load-out and subsequent hauling operations.

Future work must address these questions. As biorefinery operations mature, it is probable that an affirmative answer to questions 1 and 2 will be incorporated. The option to accumulate bale-in-rack inventory to supply the biorefinery at full capacity during weather delays (question 3) will require a significant investment in additional racks. The cost of this investment will be compared to the cost penalty incurred when the biorefinery reduces production due to a feedstock shortage.

## 5. Conclusions

A multi-bale handling unit offers an advantage for the efficient hauling of round bales. Of equal, or greater importance, is the advantage provided for efficient flow of material into, and out of, short-term storage at a biorefinery. The concept presented here is a 20-bale rack mounted on a trailer with two trailers hooked in tandem to form a 40-bale load. Two empty racks on trailers are left at an SSL for loading while a truck tractor delivers two loaded racks to the biorefinery, thus uncoupling the loading and hauling operations, and increasing the efficiency of both. The goal for unhooking two empty trailers and hooking two full trailers is 10 min. This load time equals the projected 10 min unload time at the biorefinery achieved by lifting off the two full racks and replacing them with two empties, a technology adapted from the container shipping industry.

The business plan presented here envisions that the biorefinery will own the racks and trailers and offer two contracts. One contract will hire a company to load bales into racks at the SSL and a second contract will hire truck tractors to pull the trailers. Because the biorefinery pays for all deliveries, central control is achieved. Any available truck can be dispatched to any SSL where a load is ready. As operations mature, productivity of both SSL load-out equipment and highway hauling equipment will be optimized.

The current rack design specified that the lower tier bales would be loaded from the side. No side rail was provided to keep the bales aligned during transport. Testing of this design in the rack unloader (machine to unload the 20 bales into a single line on a conveyor supplying the biorefinery) showed that the lower tier bales misaligned during transport and could not be pushed out. A new rack design was developed with a side rail to hold the bottom-tier bales in position. This new design specifies that the bottom tier bales be loaded from the ends of the racks.

A concept is presented for a bale loader which latches onto the trailer and moves bales into the bottom tier chambers of the racks. This machine will load 10 bales into the rack on the front trailer by attaching on to the front of the trailer and 10 bales into the rear trailer by attaching onto the rear. A telehandler removes bales from single-layer storage and places them in the bale loader to load the bottom tier compartments. The top tier compartments are loaded directly from the top. Expectations are that an experienced operator can average 9 loads in a 10-h workday for an average achieved productivity of 0.60. The SSL load-out cost is then estimated at 3.61 USD/Mg. If the average achieved productivity factor is 0.45, the cost is 4.81 USD/Mg, and it is 3.09 USD/Mg when the average achieved productivity factor is 0.70.

**Author Contributions:** Conceptualization, R.G., J.C., and S.C.S.; Methodology, J.C., and S.C.S.; Writing—Original Draft Preparation, J.C.; Writing—Review and Editing, R.G.; Investigation, R.G. and S.C.S.; and Project Administration, J.C., and S.C.S. All authors have read and agreed to the published version of the manuscript.

**Funding:** This research received no external funding.

**Institutional Review Board Statement:** Not applicable.

**Informed Consent Statement:** Not applicable.

**Data Availability Statement:** Not applicable.

**Conflicts of Interest:** The authors declare no conflict of interest.

## Appendix A. Cost to Operate Bale Loader

The cost to operate and own a bale loader is based on the Moffett Model M5 503 T4 with the following cost and operating factors:

Purchase Price: 51,500 USD
Salvage value: 10% of Purchase Price ($S_v = 0.1$)
Design life: 10,000 h
Annual use: 10 h/d, 6 d/wk, 48 wk/y = 2880 h/y

Interest rate: 6.25% (r = 0.0625)
Tax rate: 1%
Insurance rate: 0.80 USD/100 USD value/y

$$\text{Annual Insurance} = \frac{51,500}{100} \text{ x } 0.80 = 412 \text{ USD/y} \tag{A1}$$

Repair & maintenance factor (R/M): 2.00 USD/h
Fuel use: 3.5 L/h
Fuel cost: 0.79 USD/L
Labor cost (including benefits): 0 USD/h
(Operator of telehandler also operates SSL forklift)
Ownership cost percentage as calculated by [8]:

$$C_o = \frac{1 - S_v}{n} + \frac{(1 + S_v)r}{2} + K_2 \tag{A2}$$

where $C_o$ is the ownership cost percentage (dec), $S_v$ is the salvage value (dec), n is the expected service life (y), r is the interest rate (dec), and $K_2$ is the factor for taxes and insurance (dec). Where $K_2$ is given as: 0.01 + 0.008 = 0.018.

The expected life of the machine is:

$$n = \frac{10,000 \text{ h}}{2880 \text{ h/y}} = 3.5 \text{ y} \tag{A3}$$

The cost of ownership percentage is:

$$C_0 = \frac{1 - 0.1}{3.5} + \frac{(1 + 0.1)\,0.0625}{2} + 0.018 = 0.31$$

Annual ownership cost is 51,500 USD × 0.310 = 15,965 USD/y
Annual ownership cost (USD/h) is (15,965 USD/y)/(2880 h/y) = 5.55 USD/h
Operating Cost (USD/h) is R/M + Fuel + Labor = 2 + (3.5 × 0.79) + 0 = 4.77 USD/h
Total cost (USD/h) is Ownership + Operating = 5.55 + 4.77 = 10.32 USD/h

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
