# Peer review of "Rapid Truck Loading for Efficient Feedstock Logistics"

_agriengineering, doi:10.3390/agriengineering3020010_

Round 1

Reviewer 1 Report

This is an interesting concept paper. Design modifications to the bale rack, which address problems encountered with the previous design (which were presented in a previous publication), are described adequately. A new method for loading the bales into the bottom layer of the bale rack has been proposed (and the explanation of how this loading would occur is clear). It is not clear, however, whether a prototype bale loader has been built and tested, or whether this is solely a conceptual design. I believe it is purely conceptual, but please ensure that this is stated clearly. 

An appropriate analysis of the conceptual design has been completed. I do wonder whether the word "Results" is an appropriate section header for this section of the paper. It is not a typical paper describing experimental methods, results and discussion. Perhaps different section headings could be employed to better reflect the nature of the paper.
